# Influence of Vertebrate Excreta on Attraction, Oviposition and Development of the Asian Tiger Mosquito, *Aedes albopictus* (Diptera: Culicidae)

**DOI:** 10.3390/insects12040313

**Published:** 2021-04-01

**Authors:** R. Dulka T. Rajapaksha, Dona Pamoda W. Jayatunga, G. A. S. M. Ganehiarachchi

**Affiliations:** Department of Zoology and Environmental Management, Faculty of Science, University of Kelaniya, Kelaniya 11600, Sri Lanka; dulkatrajapaksha@gmail.com (R.D.T.R.); pamodajayatunga@gmail.com (D.P.W.J.)

**Keywords:** vertebrate excreta, stimulatory response, *Aedes albopictus*, oviposition

## Abstract

**Simple Summary:**

Commonly known as the Asian tiger mosquito, *Aedes albopictus* is a vector of dengue worldwide. Knowledge of the behavior of dengue vectors facilitates effective vector control. This is the first comprehensive analysis of selected vertebrate excreta of goat, cow and pig to identify the oviposition attraction and growth performance of *Ae. albopictus* in Sri Lanka. The current study revealed that *Ae. albopictus* gravid females are significantly attracted to goat excreta but are repelled by pig excreta. The oviposition preference was highest for the cow excreta and lowest for the pig excreta. For excreta combinations, the Cow+Goat combination increased the oviposition while the Pig+Goat combination reduced the oviposition. The oviposition preference of *Ae. albopictus* increased with the rate of fermentation. The pig excreta increased the *Ae. albopictus* larval mortality, larval and pupal duration and reduced adult fecundity, whereas the cow excreta positively affected all these aspects. Additionally, our findings suggest that a high abundance of *Ae. albopictus* in rural areas of Sri Lanka is possibly due to its oviposition attraction and the growth performance of the vertebrate excreta.

**Abstract:**

*Aedes albopictus* is an important vector of dengue worldwide. Eliminating dengue in Sri Lanka depends entirely on controlling the vector and human-vector contact. Thus, studying the bionomics and behavior of *Ae. albopictus* is paramount. The objective of this study was to evaluate the effect of the excreta of cow, goat and pig on the attraction, oviposition and development of *Ae. albopictus*. Bioassay chambers determined the mosquito stimulatory response. Ovitraps determined *Ae. albopictus* oviposition preference to excreta singly, in combination and on fermentation. The excreta effect on larval development was also determined. The results revealed that *Ae. albopictus* gravid females were significantly attracted to goat excreta but were repelled by pig excreta. The oviposition preference was highest for cow excreta and lowest for pig excreta. For excreta combinations, the Cow+Goat combination increased the oviposition while the Pig+Goat combination reduced the oviposition. The oviposition preference of *Ae. albopictus* increased with the rate of fermentation. The pig excreta increased the *Ae. albopictus* larval mortality, larval and pupal duration and reduced adult fecundity, whereas the cow excreta positively affected all these aspects. Our findings additionally suggest that a high abundance of *Ae. albopictus* in rural areas of Sri Lanka may be due to its oviposition attraction and growth performance for vertebrate excreta.

## 1. Introduction

Dengue is a hurdle to public health in Sri Lanka, causing high morbidity and mortality [1]. Two *Aedes* species that transmit the dengue virus are well established in the country; *Aedes albopictus* and *Aedes aegypti*. Both belong to family Culicidae and subfamily Culicinae [2]. In Sri Lanka, eliminating dengue depends entirely on vector control or sabotaging the human–vector contact. Therefore, studying the bionomics and the behavior of the dengue vectors facilitates effective vector control.

Oviposition site selection is critical for mosquito survival and affects the longevity of eggs, larval development and growth performance, predator avoidance, intra and interspecific interactions and offspring phenotypes and fitness [3]. Chemical cues play a crucial role in the selection of oviposition sites. The volatile chemicals from an oviposition site are sensed by mosquito olfactory receptors located on the antennae, palps, labrum and tarsi [4]. The odors released from semiochemical substances from decaying organic matter in the oviposition sites influence the mosquito egg deposition [5]. The *Ae. albopictus* follows the hypothesis of preference performance for oviposition site selection [3] as the adult female oviposits in the most suitable habitat to maximize offspring fitness while minimizing the resource-mediated interactions [6].

In a previous study, the oviposition response of *Ae. aegypti* and *Ae. albopictus* for leaves of bamboo (*Arundinaria gigantea*), white oak (*Quercus alba*), live oak (*Quercus virginiana*), pecan (*Carya illinoensis*), hackberry (*Celtis occidentalis*), red maple (*Acer rubrum*), redtop panicgrass (*Panicum rigidulum*) and harvested Bermuda grass (*Cynodon dactylon*) have been determined [7]. They reported that semiochemicals formed by the breaking down of organic matter through microbial metabolic activity caused the female *Aedes* mosquitoes to oviposit [7].

It is reported that *Ae. albopictus* is more abundant in the rural areas than in the urban areas of Sri Lanka [8]. In addition to the usual natural breeding sites, such as tree holes and rock pools and artificial freshwater collections, *Ae. albopictus* also oviposits in groundwater bodies with brackish water [9]. On the other hand, most rural areas have ample vertebrate excreta because of livestock practices such as poultry farming, dairy farming and pig farming. These vertebrate excreta make oviposition sites rich in nutrients and this may underlie the increased population size of *Ae. albopictus* mosquitoes in these areas. Therefore, this study was carried out to investigate how different vertebrate excreta affect the oviposition and growth performance of *Ae. albopictus*. Moreover, the excreta combinations were analyzed for potential synergistic effects on the oviposition of *Ae. albopictus* mosquitoes. 

## 2. Materials and Methods

### 2.1. Mosquito Rearing

The experiments were carried out during April to November 2018. The eggs of *Ae. albopictus* were obtained on paper sheets from inbred mosquito colonies at the Medical Research Institute (MRI), Sri Lanka. A pure colony consisting of only *Ae. albopictus* was maintained during the entire study period in the insectary (temperature: 27 °C; relative humidity 80%; photoperiod of 12 h) at the University of Kelaniya, Sri Lanka. For that, the egg sheets obtained were immersed in a 500 mL glass bottle filled with aged tap water. Following egg hatching, the larvae of second instar were transferred into 750 mL enamel trays covered with a net. The larvae were fed with larval food (crushed shrimp powder) until they pupated. The pupae were transferred to another tray containing aged tap water and kept inside mosquito rearing cages until adults emerged.

These adults were fed with a 10% sucrose solution for two days and then starved for 24 h prior to the experiments. On the experiment day, a blood-filled membrane was placed in the cage in a manner that the blood containing side was facing the interior of the cage. The blood-fed gravid females were collected using an aspirator. These blood-fed gravid females were used for all experiments.

### 2.2. Preparation of Excreta Infusions

The excreta solutions were prepared from three types of livestock animals: Pig (*Sus domesticus*), cow (*Bos taurus*) and goat (*Capra hircus*). Fresh feces of pigs, cows and goats were collected from a village in Mawathagama, Sri Lanka (7°52′38.72′′ N 80°42′1.23′′ E) (Gamin etrex©, Garmin Ltd., Olathe, KS, USA). These feces were air dried under the sun for 10 days. Then, the feces were finely ground and sieved through a small mesh and stored on air-tight containers [10]. The excreta infusions were prepared by mixing 10 g of crushed excreta in 200 mL of water. 

### 2.3. Screening the Attraction and Stimulatory Response of Female Mosquitoes towards the Animal Excreta for Oviposition

To evaluate the attraction and stimulatory response of ovipositing female *Ae. albopictus* towards the selected vertebrate excreta, a 4-way bioassay choice chamber was prepared. As shown in Figure 1, the bioassay setup consisted of 4 chambers, joined to the centrally placed insertion chamber with 4 hard, transparent joining tubes 9 cm × 2.5 cm (length × diameter). Three bioassay setups were set in an experiment in a random manner changing the position of the treatment. Aged tap water was used for the control chamber.

A fresh batch of 20 blood-fed gravid female *Ae. albopictus* were released into the insertion chamber of each bioassay set up at 5.00 pm, approximately one hour before starting night time. (The insectary lighting conditions did not allow dusk/dawn periods). The number of mosquitoes that had moved and settled on the different treatments were counted after the first hour (6.00 pm). This experiment was repeated for six consecutive days using same-aged fresh female batches. 

### 2.4. Effect of Different Vertebrate Excreta Infusions on the Oviposition of Aedes albopictus Mosquitoes

Three different vertebrate excreta solutions described above were used for this experiment. Aged tap water was used as the control. Four ovitraps with filter paper stripes containing each vertebrate excreta and control were placed in the experimental cage (75 cm × 45 cm × 45 cm) in a random manner. Total of 16 ovitraps were placed in a cage. This was done in separate cages for each type of excreta. Considering the fermented excreta to be microenvironments for oviposition, these ovitraps were considered as microcosms [11]. Twenty blood-fed gravid females of *Ae. albopictus* were released into each cage and allowed to stay in for five days. The eggs in the microcosms were counted every day in each ovitrap and recorded separately. The whole experiment was replicated 4 times. 

### 2.5. Effect of Vertebrate Excreta Combinations on the Oviposition Response of Aedes albopictus

In this study, four excreta combinations were used: Three combinations of two-vertebrate excreta and one combination of three-vertebrate excreta. The two-excreta combinations were Cow+Pig, Cow+Goat and Pig+Goat. For these combinations, 50% of excreta from each type were mixed. The other combination, Cow+Pig+Goat had 33% of excreta from each vertebrate combined.

To study the effect of the two-excreta combinations and the three-excreta combinations, four ovitraps for each of the combination (total of 16 ovitraps) were randomly arranged in the mosquito rearing cage and four containers of aerated distilled water were placed in the cage as the control. Then, 20 mated gravid females were released into the cage. They were allowed in the cages for 5 days. The eggs were counted every day in each ovitrap and recorded separately. This experiment was replicated 4 times. 

### 2.6. Days of Fermentation of Vertebrate Excreta and Oviposition Preference of Aedes albopictus

A series of vertebrate excreta infusions was prepared and allowed to ferment for 14, 12, 10, 8, 6, 4, 2 and 0 days (the control). Two containers of vertebrate excreta with differing days of fermentation were placed in mosquito cages, one cage for each type of excreta (total of 16 microcosms of the same excreta type within a cage). A filter paper was placed in each container as an oviposition substrate. These microcosms were exposed to the mated blood-fed gravid females of *Ae. albopictus* for oviposition. The eggs were counted after 2 days of female introduction. This experiment was replicated 4 times.

### 2.7. Effect of Different Vertebrate Excreta on the Development of Aedes albopictus

Four different microcosms were prepared from each of the three excreta types (200 mL) and placed in separate mosquito cages. Another 4 aerated water containers were also used as the controls. Twenty first instar larvae were introduced to each microcosm (altogether 16 microcosms in a cage). Three cages were set up in an experiment. Each microcosm was observed daily until the emergence of adults. Larval mortality was measured by recording the number of dead larvae. Both larval and pupal duration were measured by counting the days from first instar introduction to formation of pupae and pupal formation to emergence of adult, respectively. Mosquito fecundity was measured by the right-wing length of the adult female. For that, the emerged adults were killed using ethyl acetate. The right wing of each female adult was removed at the time of dissection and mounted on a glass microscope slide in a small drop of distilled water. The wing length was measured from the axial incision to the apical end of the wing [12] using a compound microscope (Olympus© CX21 Microscope, Tokyo, Japan) with high power (40× magnification. The wing was aligned along an ocular micrometer (1 ocular unit, 0.025 mm at 40×) and the wing length was measured to the nearest 0.01 m [13]. This experiment was replicated 3 times.

### 2.8. Data Analysis

Statistical analysis was performed using SPSS V.20. One-Way ANOVA and Tukey–Kramer HSD test were performed to determine the effect of vertebrate excreta infusions on mosquito attraction for the stimulatory response, mosquito attraction for oviposition, the effect of vertebrate excreta combinations on the oviposition of *Ae. albopictus* mosquitoes and effect of different vertebrate excreta on the development of *Ae. albopictus*. Correlation, regression analysis, One-Way ANOVA and Tukey–Kramer HSD tests were performed to determine the effect of fermenting rate of vertebrate excreta on the oviposition of *Ae. albopictus* mosquitoes. All experiments were replicated at least 3 times. All data were expressed as mean ± standard error (SE).

## 3. Results

### 3.1. Highest Stimulatory Response of Aedes albopictus towards Goat Excreta

When different excreta infusions were given in a bioassay choice chamber, gravid females of *Ae. albopictus* showed a significant stimulatory response towards the different vertebrate excreta (ANOVA, F_3, 68_ = 27.93, *p* < 0.05). It was observed that the goat excreta received significantly higher attraction, while pig excreta received the significantly lowest attraction compared to the control (Figure 2). 

### 3.2. Oviposition Preference of Aedes albopictus towards Cow Excreta

In the second experiment, gravid females of *Ae. albopictus* showed a significant oviposition response towards the different types of excreta (ANOVA, F_3, 60_ = 242.5, *p* < 0.05) compared to the control. Cow excreta received the highest number of eggs, followed by that of goat and pig, respectively (Figure 3).

### 3.3. Oviposition Preference of Aedes albopictus towards Excreta Combination of Cow+Goat

When different combinations of excreta were given in the choice test, gravid females of *Ae. albopictus* showed a significant oviposition response towards the different combinations of vertebrate excreta (ANOVA, F_4,75_ = 120.81, *p* < 0.001). The maximum oviposition response was observed in the excreta combination of Cow+Goat (Figure 4).

### 3.4. Fermented Excreta Positively Influence Oviposition of Aedes albopictus

The influence of fermentation of vertebrate excreta on the oviposition of *Ae. albopictus* was studied in this experiment. It was observed that the days of fermentation had a significantly higher effect on the oviposition of *Ae. albopictus*. The mean number of eggs was increased with the increase of fermentation days of pig excreta (Pearson’s correlation, *p* < 0.001, R^2^ = 89.7); goat excreta (Pearson’s correlation, *p* < 0.001, R^2^ = 91.1) and cow excreta (Pearson’s correlation, *p* < 0.001, R^2^ = 80.9). In all experiments, it was clearly observed that maximum number of eggs was deposited for all excreta after 12th and 14th days of fermentation. Pig (One-way ANOVA, F_7, 24_ = 38.39, *p* < 0.001); cow (One-way ANOVA, F_7, 24_ = 28.89, *p* < 0.001); goat (One-way ANOVA, F_7, 24_ = 128.48, *p* < 0.001) (Figure 5).

### 3.5. Increased Larval Mortality for Pig Excreta

Different excreta had significant effects on larval mortality. A significant difference of the mean larval mortality of *Ae. albopictus* for each of the different vertebrate excreta (One-way ANOVA, F_3, 46_ = 31.68, *p* < 0.001) was observed. The maximum percentage larval mortality was observed in the pig infusion and the minimum number of total larval mortality was observed from the cow excreta infusion (Figure 6).

### 3.6. Decreased Duration of Larvae and Pupae for All Excreta Types

All three vertebrate excreta significantly decreased the larval duration of *Ae. albopictus* (One-way ANOVA, F_3, 44_ = 117, *p* < 0.001). However, the days of larval duration for the three excreta types were not significantly different (Figure 7a). 

Pupal duration of the *Ae. albopictus* was also significantly reduced by different vertebrate excreta tested (One-Way ANOVA, F_3, 44_ = 18, *p* < 0.001). The days of pupal duration were higher for both pig and goat (statistically nonsignificant from each other) and minimal pupal duration was observed for the cow infusion (Figure 7b).

### 3.7. Increased Fecundity of Aedes albopictus to Cow Excreta

Different vertebrate excreta had a significant effect on the fecundity of adult *Ae. albopictus* (One-Way ANOVA, F_3, 46_ = 370.40, *p* < 0.001). The maximum mean wing length was observed from the cow excreta infusion and the minimum mean wing length was observed from the pig excreta infusion (Figure 8).

## 4. Discussion

Previous studies have shown vertebrate excreta as potential oviposition attractants and growth enhancers for the sand fly species *Lutzomyia longipalpis* and *Phlebotomus argentipes* (Diptera: Psychodidae) [10]. However, no such study has been conducted for mosquitoes other than different organic infusions that were shown to be oviposition attractants [14,15]. Ponnusamy et al. [7] stated that the semiochemicals produced by microbial metabolic activity in organic matter elicited the female *Aedes* mosquitoes to oviposit. In fact, the encounter of vertebrate excreta in *Ae. albopictus* oviposition sites is possible as Aedes mosquitoes breed in brackish water [9]. Considering that, this is the first comprehensive analysis of selected vertebrate excreta of goat (*Capra hircus*), cow (*Bos taurus*) and pig (*Sus domesticus*) to identify the oviposition attraction and growth performance of *Ae. albopictus* in Sri Lanka.

The feeding habits of vertebrates are different to each other. The feeding habits and physiology determine the vertebrate gut microbiota, which is a substantial component in excreta [16,17,18,19,20,21,22]. The microbial communities in the vertebrate excreta may act as food sources of mosquito larvae for growth and development. They also produce nonvolatile and volatile chemicals via decomposition of detritus material in these larval habitats [23]. Some of these compounds may serve as semiochemicals that mediate the selection of an oviposition site and are largely responsible for the spatial distribution of mosquito species in nature [23]. Moreover, some of them act as repellents for gravid female mosquitoes while some others act as attractants [24].

In the present study, it was observed that goat excreta infusion was the most attractive stimulus for *Ae. albopictus* mosquitoes, followed by that of cow and pig excreta, respectively. These results suggest chemical composition and microorganisms inhabiting in goat excreta infusion may greatly differ from the other tested excreta types, thus resulted in semiochemicals that are highly attractive for *Ae. albopictus.* The pig infusions had a deterrent effect for *Ae. albopictus* mosquitoes. It indicates that chemical composition and microorganisms inhabiting pig excreta may differ and may form semiochemicals that adversely affect *Ae. albopictus*. The goat and cow are herbivorous while the pig is omnivorous. The microbial communities within herbivorous intestines are different to those of omnivorous intestines. The microbes in the herbivorous excreta may produce semiochemicals which can act as attractants for the gravid female mosquitoes of *Ae. albopictus*. The microbial communities in the intestine of omnivorous vertebrates may produce semiochemicals which can act as a repellent for the gravid female mosquitoes of *Ae. albopictus*.

Oviposition site selection by *Ae. albopictus* follows the predictions of the preference performance hypothesis for oviposition site selection. Females allocate most of their eggs in an array of habitats that confer high offspring performance and fitness [3]. According to the optimal oviposition theory, the larval success of insects depends on the oviposition site selection by females. Females are expected to choose a site with many resources and few competitors or predators to allow the best performance for their progeny, assuming “mother knows best” [25]. 

Interestingly, the results of the effect of different vertebrate excreta on the oviposition of *Ae. albopictus* were different compared to the effect of vertebrate excreta for the stimulatory response of *Ae. albopictus*. Even though goat excreta showed the highest stimulatory response, cow excreta showed the highest oviposition response. This suggests that mosquito attraction sites do not necessarily act as mosquito oviposition sites and vice versa. Gravid female mosquitoes may lay eggs in high nutritional habitats for high offspring performance and fitness. Hence, higher number of eggs were found in cow excreta infusions. This is aligned with the preference performance hypothesis which states that female insects evolve to oviposit so that such their offspring thrive at their best [26]. Cow excreta may decompose to provide more nutrients for the neonate larvae. The minimum mosquito attraction and oviposition was observed in pig excreta infusions. This is mainly due to low stimulatory response towards the pig excreta. Moreover, the present study showed that control water is preferred for oviposition over pig excreta, indicating repellence for pig excreta. There are only low amounts of nutrients in pig excreta [27], which may be due to the low amounts of semiochemicals and nutrients produced by the specific microbial communities inhabiting the pig excreta. It is suggestive that the nutrient and mineral content, as well as the digestibility of animal excreta, greatly influence the selection of ovipositing sites by mosquitoes. 

In the present study, different combinations of selected vertebrate excreta were also evaluated for the oviposition preference of *Ae. albopictus*. Even though cow excreta showed the highest oviposition responses, it received a substantially lower number of eggs when combined with pig excreta, suggesting highly repellent characteristics of pig excreta. However, this did not happen when goat excreta were combined with cow excreta. Instead, Cow+Goat excreta showed enhanced oviposition of *Ae. albopictus*, possibly due to formation of a highly favorable semiochemical for the mosquito offspring. This combination may decompose rapidly and provide nutrients than excreta per se. The combination of cow excreta and goat excreta showed the highest oviposition. Still, it cannot be considered a synergistic increase compared to the oviposition preference per se. These results are in line with a previous study in which some combinations of leaves act synergistically, resulting in a higher total yield of mosquitoes while certain leaf combinations act antagonistically for *Ae. albopictus* [28]. 

When cow and goat excreta were combined with pig excreta (Cow+Pig+Goat), a minimum oviposition response was shown. Furthermore, it was observed oviposition of *Ae. albopictus* was reduced in combinations which include pig excreta. This indicates the chemical compounds produced from pig excreta can reduce the oviposition-enhancing features of both cow and goat excreta by damaging their microbial community. On the other hand, the least nutrients provided by the pig excreta do not support the selection of the oviposition site by *Ae. albopictus*.

This study further revealed duration of fermenting of different vertebrate excreta had a significant effect on the oviposition preference of gravid female *Ae. albopictus* mosquitoes. In all experiments, it was clearly observed that maximum numbers of eggs were oviposited after 12th and 14th days of fermentation. The fermentation process accelerated when the number of fermenting days increased. Therefore, the present study indicated the gravid female *Ae. albopictus* have a greater affinity to the oviposition sites that encompass highly fermented vertebrate excreta. 

Cow excreta hold a large amount of water, facilitating the bacterial colonies to decompose down and give off their nutrients slowly [5]. However, it is also reported that cow excreta decompose more rapidly than goat excreta [29]. It may be due to the composition of food (grass). This may be a possible reason for the higher number of eggs in the cow excreta infusions. Conversely, the fewer number of eggs in the pig excreta infusions suggests the oviposition choice of gravid females of *Ae. albopictus* is low on pig excreta. This observation suggests that pig excreta may ferment and decay slowly, providing only low amounts of nutrients into pig excreta infusions. Moreover, compared to cow and goat excreta, which have a pH of 8.5−9.0 [30], pig excreta are acidic, with a pH of 6.85 [31]. On the other hand, the optimal pH for *Aedes* larvae is reported as 7.4 [32]. Therefore, the slight acidity of pig excreta could be a reason for it to receive the lowest number of eggs in the present study. Additionally, it should be of note that in Sri Lanka, rural pig farming does not make use of antibiotics and other growth hormones; therefore, these factors do not underlie the reduced oviposition for pig excreta. 

The immature stages of container-dwelling mosquitoes are placed in their habitat by ovipositing females that lay their eggs in habitats that maximize the performance of their offspring [6]. In the present study, it was shown that growth performance of *Ae. albopictus* varied among different vertebrate excreta.

The current study revealed a significant difference between total larval mortality of *Ae. albopictus* with the different vertebrate excreta tested. The minimum larval mortality was observed in cow excreta infusions, while maximum larval mortality was recorded for the pig excreta infusion. The experimental values of the larval mortality for the control (aged tap water) were fairly high. Even though aged tap water was used as the control, maybe because it was first instar larvae that were introduced, the agitation caused during handling could have been fatal to the larvae. On the other hand, because it is not a natural habitat but artificially induced laboratory conditions, a base level of larval mortality can be expected.

The chemical substances present in pig excreta may adversely affect the survival of *Ae. albopictus* larvae. A study inferenced that rabbit excreta enhance the survival of the *Phlebotomus* sand fly [10]. Only a few experiments have been conducted to assess the excreta-based performances of mosquitoes or other medically important insects. 

The vertebrate excreta tested here had significant effects on the larval duration times. Minimum larval duration was recorded for goat excreta infusion, while maximum larval duration was recorded for both cow and pig excreta infusions. These results, while seemingly counterintuitive, require further research and confirmation. 

Fecundity should be critically defined in female mosquitoes as they are responsible for producing eggs and maintaining the population size. The current study revealed a significant difference between the wing length measurements and the type of vertebrate excreta. The adult mosquitoes that emerged from cow excreta infusion had a high wing length, indicating the influence of cow excreta for the greater fecundity of *Ae. albopictus*. It is mainly because of the rapidly decomposing cow excreta that produce nutrient-rich microcosms. The nutrients may enhance the growth of the larvae, resulting in an adult mosquito population with a larger body size.

## 5. Conclusions

In the current study, *Ae. albopictus* showed different oviposition preferences for different vertebrate excreta, possibly due to the different inherent microbial communities in different excreta. Additionally, the vertebrate excreta have a significant effect on the performance of *Ae. albopictus*, supporting the preference performance hypothesis. The reasons for such phenomena may be the nutritive basis of the oviposited microcosms and the attraction for the oviposition. On the other hand, there may be a plethora of reasons for the high abundance of *Ae. albopictus* in rural areas of Sri Lanka. Our findings additionally suggest that oviposition attraction and growth performance of mosquitoes for vertebrate excreta may be one of the reasons. However, further studies are warranted to confirm this and to identify the microbial communities and volatile and nonvolatile chemicals associated with cow, goat and pig excreta according to their feeding habits. In addition, the types of nutrients present in the selected vertebrate excreta should be studied to obtain a real insight of their influence.

## Figures and Tables

**Figure 1 insects-12-00313-f001:**
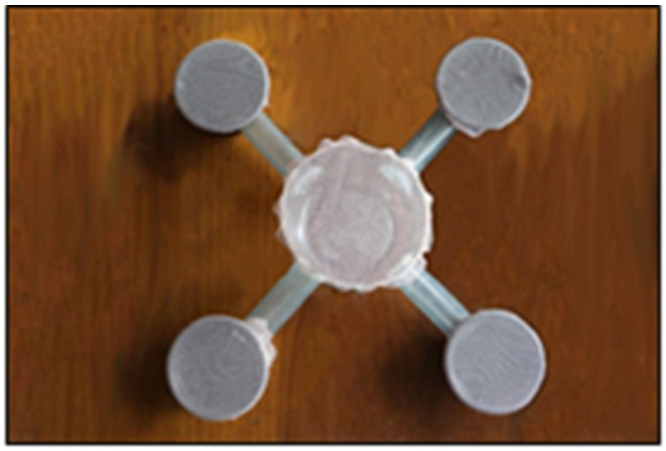
A four-way bioassay setup.

**Figure 2 insects-12-00313-f002:**
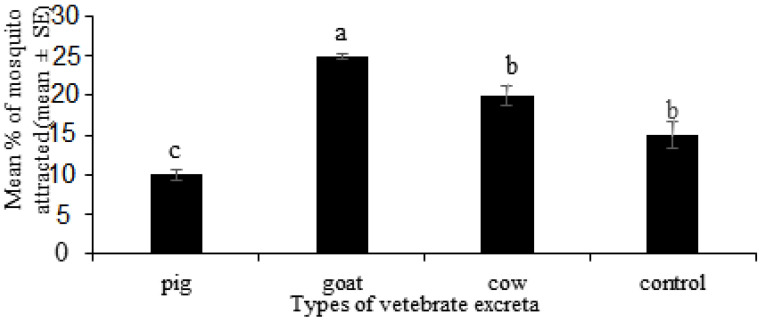
Stimulatory response of *Aedes albopictus* towards different types of vertebrate excreta: Pig (*Sus domesticus*)*,* goat (*Capra hircus*) and cow (*Bos taurus*). A four-way bioassay choice chamber was used to evaluate the attraction and stimulatory response of *Ae. albopictus* towards the selected vertebrate excreta. The mean number of mosquitoes moving towards the different excreta was counted after the first hour of exposure. Data are expressed as the mean number of mosquitoes attracted ± SE from six independent experiments (*n* = 6). Bars with different letters are significantly different (*p* < 0.05). Statistical analysis was performed using SPSS V.20.

**Figure 3 insects-12-00313-f003:**
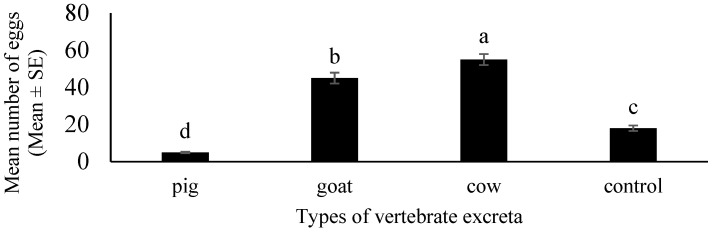
The effect of different vertebrate excreta on oviposition preference of *Aedes albopictus*. Oviposition preference was quantified by counting the mean numbers of eggs laid by the mosquitoes in ovitraps containing control water and different types of vertebrate excreta: Pig (*Sus domesticus*), goat (*Capra hircus*) and cow (*Bos taurus*). Data are expressed as mean number of eggs ± SE from four independent experiments (*n* = 4). Bars with different letters are significantly different (*p* < 0.05). Statistical analysis was performed using SPSS V.20.

**Figure 4 insects-12-00313-f004:**
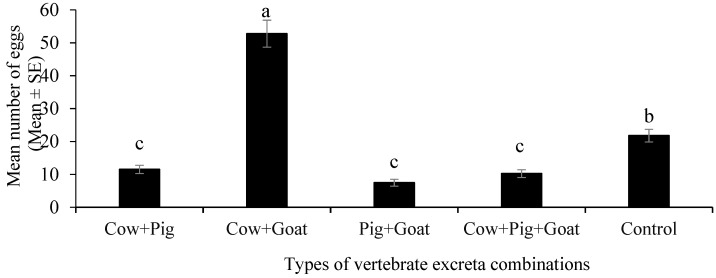
The effect of different vertebrate excreta combinations on oviposition preference of *Aedes albopictus*. Oviposition preference was quantified by counting the mean numbers of eggs laid by the mosquitoes in ovitraps containing control water and different types of vertebrate excreta combinations of pig (*Sus domesticus*), goat (*Capra hircus*) and cow (*Bos taurus*): Cow+Pig, Cow+Goat, Pig+Goat and Cow+Pig+Goat. Data are expressed as mean number of eggs ± SE from four independent experiments (*n* = 4). Bars with different letters are significantly different (*p* < 0.05). Statistical analysis was performed using SPSS V.20.

**Figure 5 insects-12-00313-f005:**
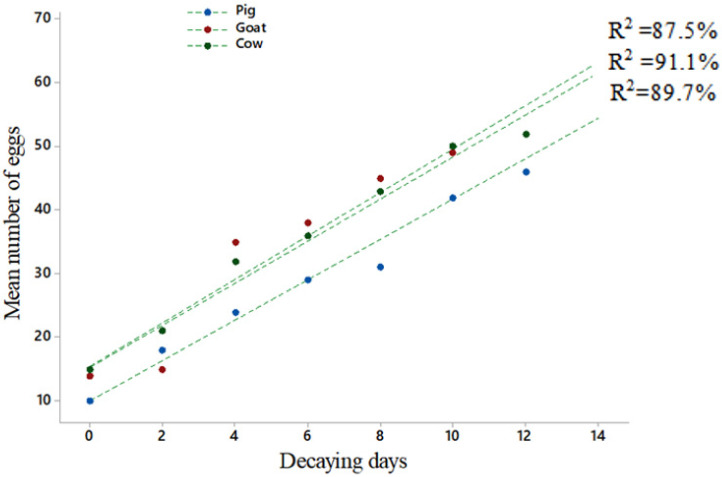
The effect of fermentation of vertebrate excreta on oviposition preference of *Aedes albopictus*. Oviposition preference was quantified by counting the mean numbers of eggs laid by the mosquitoes in the filter paper placed in each microcosm containing vertebrate excreta of pig (*Sus domesticus*), goat (*Capra hircus*) and cow (*Bos taurus*), fermented for 14, 12, 10, 8, 6, 4, 2 and 0 days. R^2^ = coefficient of determination. Statistical analysis was performed using SPSS V.20.

**Figure 6 insects-12-00313-f006:**
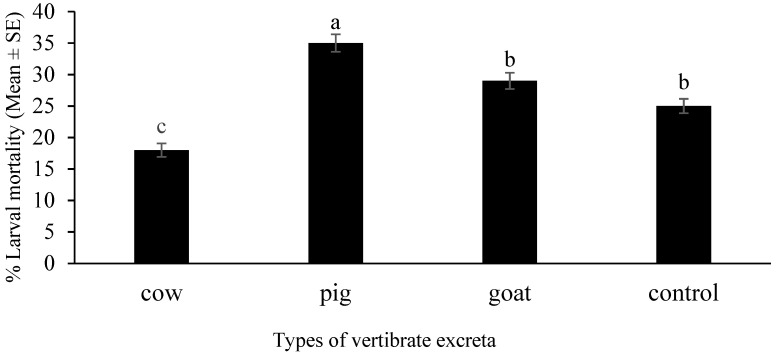
The effect of different vertebrate excreta on larval mortality of *Aedes albopictus*. Larval mortality was measured by recording the number of dead larvae from the first instar larvae introduced to microcosms prepared from excreta of pig (*Sus domesticus*), goat (*Capra hircus*) and cow (*Bos taurus*). Data are expressed as larval mortality ± SE from three independent experiments (*n* = 3). Bars with different letters are significantly different (*p* < 0.05). Statistical analysis was performed using SPSS V.20.

**Figure 7 insects-12-00313-f007:**
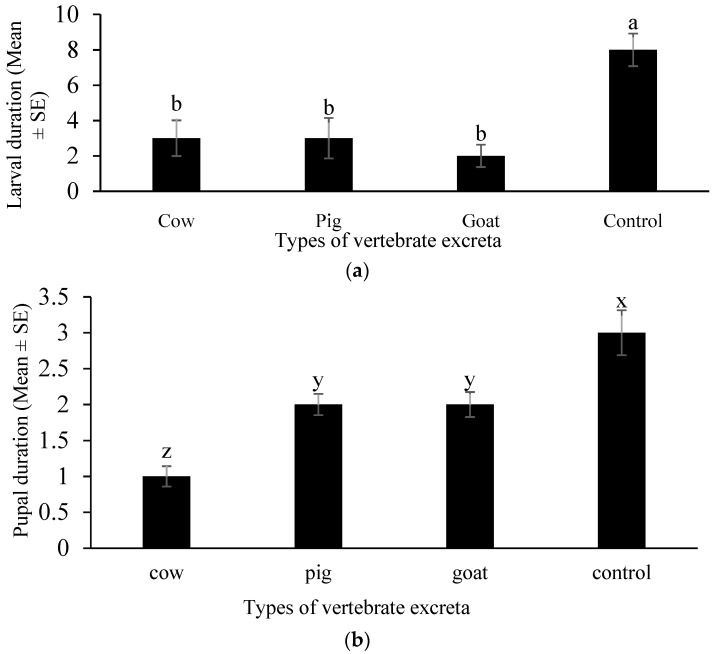
Life stage durations of *Aedes albopictus* towards different vertebrate excreta. (**a**) Larvae were introduced separately (*n* = 720 for each excreta type) for cups containing excreta of pig (*Sus domesticus*)*,* goat (*Capra hircus*) and cow (*Bos taurus*). The days of duration of larvae was recorded. (**b**) For the same batch, upon pupation, the days of pupal duration were recorded. Bars with different letters are significantly different (*p* < 0.05). Statistical analysis was performed using SPSS V.20.

**Figure 8 insects-12-00313-f008:**
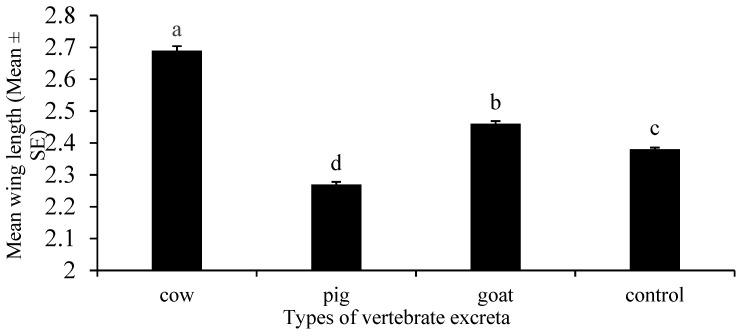
The effect of different vertebrate excreta on fecundity of *Aedes albopictus*. Fecundity was determined by measuring the mean wing length of adult female mosquitoes emerged from microcosms prepared from each excreta type: Pig (*Sus domesticus*), goat (*Capra hircus*) and cow (*Bos taurus*). Data are expressed as mean wing length ± SE from three independent experiments (*n* = 3). Bars with different letters are significantly different (*p* < 0.05). Statistical analysis was performed using SPSS V.20.

## Data Availability

All datasets supporting the conclusion of this article are included in the article. Data will not be shared in any other source.

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
