# Peer review of "Influence of Vertebrate Excreta on Attraction, Oviposition and Development of the Asian Tiger Mosquito, Aedes albopictus (Diptera: Culicidae)"

_insects, 2021, doi:10.3390/insects12040313_

Round 1

Reviewer 1 Report

This manuscript present a choice study of different animal excreta as substrate for oviposition of Aedes albopictus females. Three animal excreta are compared to water; while the experimental design is good, it seems weak to support some of the conclusions.

An interesting outcome is the low preference for pig excreta, which could be explained as suggested by the authors by a lack of nutrients; however this is not backed up by references. I recommend the authors to dig a bit further, for eg. on the acidity level or on the potential presence of antibiotics (eg. ivermectine which is deadly to female mosquitoes). It is important that antibiotics be discussed as they can be found in the excreta.

The conclusion that this finding can explain why this mosquito species is present in the rural area is not appropriate. They are of course many other reasons, and many other ovipositing sources for these mosquitoes. The authors must also discuss how likely are excreta to be found in oviposition sites of Aedes albopictus, which very rarely is found laying eggs in puddles, unlike Anopheles. This argument is crucial to demonstrate the relevance of the study and the conclusions made.

Please also review abstract accordingly.

Main comments:

L63: check the wording “the more…than”

L72: This section doesnt say where the study area is, this indication is given in the following one. This section could rather be called “mosquito colony and rearing”

L74: what’s a pure colony?

L76: The coordinates of the university are not necessary. It’d be more interesting to give the location where the initial population was collected. where did the strain at MRI come from?

Please give information on the rearing methodology and insectary conditions (temperature, RH, photoperiod)

L79: were they from the same farm or did you sample in several different farms? what can you say about the antibiotic treatment these animals usually receive?

L83: can you precise how many days of fermentation for all experiments except 2.6 ? This is important to analyse the experiment of larval growth

L93:

- (5:00pm) please mention if your insectary light conditions allow a dusk/dawn period, and when does it start (how long after 5pm?)?

-“that moved”

L94: during your hour of observation: were the mosquito able to move back after visiting one ovitrap? It doesn’t seem to me that they were trapped in the ovitrap (by sticky plastic sheet for eg.); how can you know if they directly went to the trap or if they could sense and leave?

L95: were they repeated with a new batch of females?

L101: can you describe how were the ovitraps setup? (using paper/wood?)

L103: what is the size of the cage? it is important to precise how large was the cage and the distance between the cups (was there sufficient distance to avoid odor confusion?). This is important for the repeatability of your work.

L105: were the eggs removed every day?

L106: “4 times (4 days)”: not clear, were the cages left for 5 days and the oviposition counted every day? why 4 days mentioned here?

L121: did you actually measure the rate of fermentation or are you only talking about the duration?

L122: can you really call it a microcosm?

L123: why « separately »?

L128: first mention of blocks: what are they?

L129: Please precise quantity of liquid? larval density?

L131: not clear, was there one block for each animal excreta? this initial description could be made cleared

L135: why fecundity? just mention mosquito size please.

L157-8: did you observe female behaviour during the one hour of the experiment? 

it would be important to report if they have tried going to the pig trap and were repulsed at close contact or were they immediately more attracted by the goat and cow traps?

Figure 2: this graph is not well fitted for a choice experiment. i recommend using mean% of females attracted towards each trap.

Was the trend similar in all the replicates? (ie. always goat scoring more females, followed by cow?)

L162 (and for each legend): no need to repeat species name here in my opinion

L163: did you observe if the female oviposited? if not you may only mention attraction

L164: “moving”

L172: Please correct: control/water comes before pig, this is an important result.

L196: “higher”

L206: Please use “size” not “fecundity”. fecundity = number of eggs

L207: please rephrase: the difference is not between mortality and excreta

Figure 6:

- it is surprising that the larvae don’t do worst in the control (if it was only clear water) than with excreta…this is showing that excreta alone is not sufficient for development. It may only have a role of attraction

-I dont understand the numbers: there were initially 20 first instar larvae per cup; what’s the unit used here for mortality? is it a percentage? if so please precise it please, as it is confusing

It is hard to review the relevance of these data without understanding the actual numbers.

L227: 2 to 3 days larval duration is very low for albopictus! how is that possible? This needs to be checked and explained.

Figure 8: as wing length of females and males are quite different in size for albopictus: which sex did you use here? if they were mixed then please separate them.

L263: another very important parameter to consider is the antibiotic treatments, eg ivermectin. Can you please comment on the usual use of antibiotics for these 3 animals in Sri Lanka

L273: i dont think you can say that as pig excreta was less attractive that water.

L279: and pigs are usually among the animals most treated with antibiotics

L292-4: please rephrase, not very clear

L30&-3: is this an hypothesis or is there reference to back this up?

L311-2: or it shows the highly repellent characteristics of pig excreta.. the fact that water is preferred over pig excreta is not discussed, however this is an interesting outcome that deserve explanation

L314-6: you would need to do a choice experiment to prove that. To me it mostly shows that pig excreta is repulsive

L324-5: the sentence is not clear. Is there a reference indicating the low nutrients in the pig excreta or is it an hypothesis?

to document this it would have been interesting to compare a 14days-fermented-pig trap vs a 2days-fermented-cow trap….

L335: “more rapidly”, i could not see this data about the speed of decomposition, are you referring to personal observation?

L338-9: this sentence is not needed as it is obvious

L340: there could be several other explanations such as acidity and antibiotics…

L342: please review this sentence

L344: i am surprised by the larval growth in the controls, this result in my opinion lowers the impact of your observations. This in any case should be discussed.

L350-3: these sentences are not discussed and seem a bit out of place

L356-7: due to a high initial mortality these results may also have a low value of interpretation…

L358: although fecundity may be related to wing size, you can not discuss on fecundity but only on a proxy of fecundity. Please only mention adult size here, and indicate if you are discussing females or males.

L372-4: I don’t agree with this conclusion, you would need to compare to different types of usual oviposition substrates for that.

You also need to comment on the relevance of excreta as substrate for albopictus oviposition sites. Are they field observations of aedes larvae on the ground? to my knowledge this is quite rare for aedes; have this been documented in the rural areas?

L378: “conducted field survey” and experimental activities?

Author Response

Reviewer 1

This manuscript present a choice study of different animal excreta as substrate for oviposition of Aedes albopictus females. Three animal excreta are compared to water; while the experimental design is good, it seems weak to support some of the conclusions.

An interesting outcome is the low preference for pig excreta, which could be explained as suggested by the authors by a lack of nutrients; however this is not backed up by references. I recommend the authors to dig a bit further, for eg. on the acidity level or on the potential presence of antibiotics (eg. ivermectine which is deadly to female mosquitoes). It is important that antibiotics be discussed as they can be found in the excreta.

The conclusion that this finding can explain why this mosquito species is present in the rural area is not appropriate. They are of course many other reasons, and many other ovipositing sources for these mosquitoes. The authors must also discuss how likely are excreta to be found in oviposition sites of Aedes albopictus, which very rarely is found laying eggs in puddles, unlike Anopheles. This argument is crucial to demonstrate the relevance of the study and the conclusions made.

Please also review abstract accordingly.

Main comments:

  1. L63: check the wording “the more…than”

                                                Re-worded the sentence.

  1. L72: This section does not say where the study area is, this indication is given in the following one. This section could rather be called “mosquito colony and rearing”

                                                Changed the section heading as advised.

  1. L74: What’s a pure colony?

                                                A pure colony consists of only one species exclusively. In this case, it was a pure colony of only Ae. albopictus. It was mentioned in the L75.

  1. L76: The coordinates of the university are not necessary. It’d be more interesting to give the location where the initial population was collected. where did the strain at MRI come from?

                                                As advised, omitted the coordinates of the university, where the study was carried out. The eggs were obtained from MRI from their inbred colonies; therefore, it is not possible to trace back from where the initial population has been collected.

  1. Please give information on the rearing methodology and insectary conditions (temperature, RH, photoperiod)

                                                Information on the rearing methodology and insectary conditions were included in L77-87.

  1. L79: were they from the same farm or did you sample in several different farms? what can you say about the antibiotic treatment these animals usually receive?

                                                The excreta samples were obtained from several different farms. As these farms were domesticated and not of commercial scale, the animals had not been given any antibiotic treatment.

  1. L83: can you precise how many days of fermentation for all experiments except 2.6 ? This is important to analyse the experiment of larval growth.

                                                There was no fermentation involved in the rest of the experiments except 2.6. the section. Therefore, fermenting part was not described under the section 2.2, the preparation of excreta infusions.

  1. L93: (5:00pm) please mention if your insectary light conditions allow a dusk/dawn period, and when does it start (how long after 5pm?)?

-“that moved”

                                                The insectary light conditions did not allow a dusk/dawn period and this was mentioned in L94. However, the insectary had a photoperiod of 12 hours and it was mentioned in L-77.

  1. L106: during your hour of observation: were the mosquito able to move back after visiting one ovitrap? It doesn’t seem to me that they were trapped in the ovitrap (by sticky plastic sheet for eg.); how can you know if they directly went to the trap or if they could sense and leave?

                                                Because of this reason of mosquitoes going back and forth, the number of mosquitoes in each type of excreta was counted only after a one hour of settling time. The mosquitoes were introduced at 5.00 pm and the mosquito counts at each excreta was done at 6.00 pm. This was mentioned more clearly in L106-107.        

  1. L108: were they repeated with a new batch of females?

                                                Yes, the experiment was repeated with a fresh batch of females and this was included in L108.

  1. L105: can you describe how were the ovitraps setup? (using paper/wood?)

                                                The ovitaps used contained filter paper stripes and this was mentioned in L115-116.

  1. L103: what is the size of the cage? it is important to precise how large was the cage and the distance between the cups (was there sufficient distance to avoid odor confusion?). This is important for the repeatability of your work.

                                                Included the size of the cage and the distance between the cups.

  1. L105: were the eggs removed every day?

                                                No, the total number of eggs existed each day was counted to see any daily changes. However, for calculations, the total no of eggs at the 5 th day was used to calculate the mean no. of egges from all 4 experiments replicated.

  1. L106: “4 times (4 days)”: not clear, were the cages left for 5 days and the oviposition counted every day? why 4 days mentioned here?

                                                4 days were mentioned as a mistake. Edited it accordingly in L18-21                                                                                          

  1. L121: did you actually measure the rate of fermentation or are you only talking about the duration

                                                It is about duration. Because we used the same amount of excreta fermented for different time periods such as 2 , 4, 6, 8, 10, 12 and 14 days. We determined the mosquito preference to oviposit in excreta with differing days of fermentation. The L-136-138 were edited accordingly for clarity.

  1. L122: can you really call it a microcosm?

                                                A microcosm is a miniature environment that aims to represent a larger one. In the present study, we considered the setups containing fermented excreta made to oviposit as microcosms. This was mentioned in L-119-120.                                          

  1. L140: why « separately »?

                                                Separate cages were used for each excreta type because the focus was to arrive at the preferred level of fermentation (day wise) within each type of excreta. This is now mentioned in the L141.

  1. L147: first mention of blocks: what are they?

                                                What we meant by “blocks” were same as the mosquito cage. However, it was suddenly introduced word, so, we omitted that word and continued referring as “cage”. Eg. L148.

  1. L129: Please precise quantity of liquid? larval density?

                                                The quantity of excreta liquid used throughout the study was 200 mL and this s mentioned in L-95. However, mentioned this again and also the 20 first instar larvae introduction in L-148-149.                   

=

  1. L131: not clear, was there one block for each animal excreta? this initial description could be made cleared.

                                                Yes, please refer to the answer of the comment no. 18.

  1. L154-155: why fecundity? just mention mosquito size please.

                                                Mosquito size measured in terms of the female right wing length is widely recognized as a measure of mosquito fecundity. References are added in the L-154,155.

  1. L157-8: did you observe female behaviour during the one hour of the experiment? 

                                                Female behaviour was not observed during the whole one hour of the experiment. Instead, after introducing the females, the mosquito counts at each of the excreta were obtained after one hour. This was mentioned more clearly in L-95, 96.

  1. 2it would be important to report if they have tried going to the pig trap and were repulsed at close contact or were they immediately more attracted by the goat and cow traps?

                                                Please refer the above answer for comment 22.

  1. Figure 2: this graph is not well fitted for a choice experiment. i recommend using mean% of females attracted towards each trap.

                                                In the figure 2, the Y axis was changed to mean percentage of mosquitoes attracted.

  1. Was the trend similar in all the replicates? (ie. always goat scoring more females, followed by cow?)

                                                Yes, in all replicates the got excreta scored more females than cow excreta. This can be seen by the Figure 2 itself, as mean % of females attracted to goat and cow excreta are significantly different as denoted by the symbols a and b (Also, their error bars do not overlap).

  1. L162 (and for each legend): no need to repeat species name here in my opinion.

                                                The species names were repeated in the figures because the general norm is that the figures should stand alone themselves. However, if reviewers still think it is not necessary, the authors can omit the names accordingly in the next revision.

  1. L163: did you observe if the female oviposited? if not you may only mention attraction.

                                                This experiment determined only the attraction of the mosquitoes to different excreta, therefore, removed the words “ ovipositing female..”

  1. L164: “moving”

                                                Corrected the word to moving.

  1. L172: Please correct: control/water comes before pig, this is an important result.

                                                Included the result of control water before the result for pig excreta.

  1. L196: “higher”

                                                Edited as suggested.

  1. L206: Please use “size” not “fecundity”. fecundity = number of eggs

                                                As mentioned before, mosquito size measured in terms of the female right wing length is widely recognized as a measure of mosquito fecundity. References are added in the L-154,155.

  1. L226: please rephrase: the difference is not between mortality and excreta.

                                                The sentence was modified in a manner it gave the meaning that a significant difference of the mean larval mortality of Ae. albopictus for each of the different vertebrate excreta was observed. -226-228.

  1. Figure 6:- it is surprising that the larvae don’t do worst in the control (if it was only clear water) than with excreta…this is showing that excreta alone is not sufficient for development. It may only have a role of attraction.

                                                In the figure 6, larvae do not do worst in the control, which is aged tap water. The results resemble that of oviposition results because the lowest % larval mortality has been reported for cow excreta. Percentage larval mortality increases respectively in the control, goat and pig. However, the result is not significantly different between the goat and control and the pig and the control. It can clearly be seen that cow excreta decreases larval mortality and pig excreta increases larval mortality compared to the control.

-I dont understand the numbers: there were initially 20 first instar larvae per cup; what’s the unit used here for mortality? is it a percentage? if so please precise it please, as it is confusing

It is hard to review the relevance of these data without understanding the actual numbers.

                                                For each cup 20 first instar larvae were introduced and there were 4 such cups at each time of doing the 4 replicates. Therefore, the total number one larvae considered per one experiment was 80. However, the Figure 6 was redrawn using % larval mortality in the y axis.

  1. L370: 2 to 3 days larval duration is very low for albopictus! how is that possible? This needs to be checked and explained.

                                                The approximate days of larval duration for Ae. albopictus is 5-10 days for artificial containers (Gomes et al., 1995). The decrease of larval duration is possible when the larval medium has high nutrients so that, more mosquito cycles occur with the favorable conditions. This was further mentioned in the discussion in L370 onwards.

  1. Figure 8: as wing length of females and males are quite different in size for albopictus: which sex did you use here? if they were mixed then please separate them.

                                                Female mosquitoes were used for this purpose. This was mentioned in L-137, 138 and L-250.

  1. L263: another very important parameter to consider is the antibiotic treatments, eg ivermectin. Can you please comment on the usual use of antibiotics for these 3 animals in Sri Lanka.

                                                In Sri Lanka, domestic livestock farms do not make use of antibiotic treatments. This was mentioned in L-374-376.

  1. L294: i dont think you can say that as pig excreta was less attractive than water.

                                                Changed the wording in L-294.

  1. L279: and pigs are usually among the animals most treated with antibiotics.

                                                Please see the answer for the comment no. 36.

  1. L313-5: please rephrase, not very clear

                                                Re-worded the L313-315.

  1. L320&-1: is this an hypothesis or is there reference to back this up?

                                                As already mentioned in the L320-321, preference performance is a hypothesis and a reference is already provided for that.

  1. L326-328: or it shows the highly repellent characteristics of pig excreta.. the fact that water is preferred over pig excreta is not discussed, however this is an interesting outcome that deserve explanation.

                                                This was included in the L326-328.                                           

  1. L334-5: you would need to do a choice experiment to prove that. To me it mostly shows that pig excreta is repulsive.

                                                Edited the sentence that mentioned about attraction.

  1. L324-5: the sentence is not clear. Is there a reference indicating the low nutrients in the pig excreta or is it an hypothesis?

                                                There is a reference stating the low nutrient levels of pig excreta compared to cow excreta. It was included in L324-325.

  1. to document this it would have been interesting to compare a 14days-fermented-pig trap vs a 2days-fermented-cow trap….

  1. L358: “more rapidly”, i could not see this data about the speed of decomposition, are you referring to personal observation?

                                                The authors agree that current study did not determine the speed of decomposition of excreta. However, this is a reported finding, and a reference was provided for that.                            

  1. L360-1: this sentence is not needed as it is obvious.

                                                The sentence was omitted.

  1. L366-370: there could be several other explanations such as acidity and antibiotics…

                                                The effect of acidity was used to explain the results here as advised.

  1. L364: please review this sentence.

                                                Edited the sentence as advised.

  1. L375: i am surprised by the larval growth in the controls, this result in my opinion lowers the impact of your observations. This in any case should be discussed.

                                                In the larval mortality experiment, the larval mortality for goat excreta was comparable to that of the control. However, the larval mortality for pig was significantly higher than the control and it was significantly lower for the cow excreta. The authors admit that the values for control were fairly high. Even though aged tap water was used as the control, may be because it was first instar larvae that were introduced, the agitation caused during transport could have been mortal to the larvae.  On the other hand, because it is not natural habitats but the laboratory conditions, a base level of larval mortality should be expected. This is mentioned in L378-382.                      

  1. L350-3: these sentences are not discussed and seem a bit out of place

                                                A phrase of the last sentence in the paragraph was deleted L-381-382.

  1. L356-7: due to a high initial mortality these results may also have a low value of interpretation…

                                                Please see the answer for the comment 49. Thank you.

  1. L358: although fecundity may be related to wing size, you can not discuss on fecundity but only on a proxy of fecundity. Please only mention adult size here, and indicate if you are discussing females or males.

                                                Please see the answer for comment no.21

  1. L407-408: I don’t agree with this conclusion, you would need to compare to different types of usual oviposition substrates for that.

                                                The authors respect the point of view of the reviewer. The authors also agree to that. However, there was a change made in L-407-408 that increase of larval performance and decrease of larval and pupal duration leading increased mosquito cycles may be one of the reasons behind the high abundance of Ae. albopictus in rural areas of Sri Lanka.

  1. You also need to comment on the relevance of excreta as substrate for albopictus oviposition sites. Are they field observations of aedes larvae on the ground? to my knowledge this is quite rare for aedes; have this been documented in the rural areas?

  1. L378: “conducted field survey” and experimental activities?

                                                Edited as conducted field work and experimental data collection.

Reviewer 2 Report

I found this to be an interesting paper.  The study is well-designed and provides insights into the effects of interactions between livestock animals and Aedes albopictus.  My main recommendation is that the authors bolster their discussion with some citation of papers that analyze microbiota and chemical makeup of excreta of the three animal species.  For example:

Ozutsumi, Y., Hayashi, H., Sakamoto, M., Itabashi, H. and Benno, Y., 2005. Culture-independent analysis of fecal microbiota in cattle. Bioscience, biotechnology, and biochemistry69(9), pp.1793-1797.

Tang, M.T., Han, H., Yu, Z., Tsuruta, T. and Nishino, N., 2017. Variability, stability, and resilience of fecal microbiota in dairy cows fed whole crop corn silage. Applied microbiology and biotechnology101(16), pp.6355-6364.

Li, H., Li, R., Chen, H., Gao, J., Wang, Y., Zhang, Y. and Qi, Z., 2020. Effect of different seasons (spring vs summer) on the microbiota diversity in the feces of dairy cows. International journal of biometeorology64(3), pp.345-354.

Jesús-Laboy, D., Kassandra, M., Godoy-Vitorino, F., Piceno, Y.M., Tom, L.M., Pantoja-Feliciano, I.G., Rivera-Rivera, M.J., Andersen, G.L. and Domínguez-Bello, M.G., 2012. Comparison of the fecal microbiota in feral and domestic goats. Genes3(1), pp.1-18.

O’Donnell, M.M., Harris, H.M., Ross, R.P. and O'Toole, P.W., 2017. Core fecal microbiota of domesticated herbivorous ruminant, hindgut fermenters, and monogastric animals. Microbiologyopen6(5), p.e00509.

Kubasova, T., Davidova-Gerzova, L., Babak, V., Cejkova, D., Montagne, L., Le-Floc'h, N. and Rychlik, I., 2018. Effects of host genetics and environmental conditions on fecal microbiota composition of pigs. PLoS One13(8), p.e0201901.

Correa-Fiz, F., Blanco-Fuertes, M., Navas, M.J., Lacasta, A., Bishop, R.P., Githaka, N., Onzere, C., Le Potier, M.F., Almagro-Delgado, V., Martinez, J. and Aragon, V., 2019. Comparative analysis of the fecal microbiota from different species of domesticated and wild suids. Scientific reports9(1), pp.1-15.

Other Comments:

Figure 5 and associated text:  It would be helpful to the reader if the authors would perform a simple linear regression analysis on these data, deriving one equation for each kind of excreta (cow, goat, pig) and then comparing the regressions statistically (MANOVA, perhaps).  As it stands, we see the interaction of different excreta at different days.  However, a regression analysis would reveal whether one kind of excreta differs from another over the entire time period, i.e., which one gave a faster oviposition response (it might be that none of them did but we do not know that).

The sentence that starts in Line 206 and runs until Line 208 does not make sense.  "It was observed that a significant difference between the mean larval mortality of Ae. albopictus and different vertebrate excreta (One-way ANOVA, F3,46= 31.68, P=0.00)."  Change this to read, "A significant difference between the mean larval mortality of Ae. albopictus and different vertebrate excreta was observed (One-way ANOVA, F3,46= 31.68, P=0.00)."

Again, Lines 208-210:  "Maximum number of total larval mortality was observed in the pig infusion and the minimum number of total larval mortality was observed from the cow excreta infusion (Figure 6)."  Change to read:  "Maximum larval mortality was observed in the pig infusion and minimum larval mortality was observed from the cow excreta infusion (Figure 6)."

Something else:  In all figure captions the authors repeat what software they used for data analysis.  The reader already was told in the Materials and Methods section - they do not need to be retold time and time again.

Author Response

Reviewer 2

I found this to be an interesting paper.  The study is well-designed and provides insights into the effects of interactions between livestock animals and Aedes albopictus

  1. My main recommendation is that the authors bolster their discussion with some citation of papers that analyze microbiota and chemical makeup of excreta of the three animal species.  For example:

Ozutsumi, Y., Hayashi, H., Sakamoto, M., Itabashi, H. and Benno, Y., 2005. Culture-independent analysis of fecal microbiota in cattle. Bioscience, biotechnology, and biochemistry69(9), pp.1793-1797.

Tang, M.T., Han, H., Yu, Z., Tsuruta, T. and Nishino, N., 2017. Variability, stability, and resilience of fecal microbiota in dairy cows fed whole crop corn silage. Applied microbiology and biotechnology101(16), pp.6355-6364.

Li, H., Li, R., Chen, H., Gao, J., Wang, Y., Zhang, Y. and Qi, Z., 2020. Effect of different seasons (spring vs summer) on the microbiota diversity in the feces of dairy cows. International journal of biometeorology64(3), pp.345-354.

Jesús-Laboy, D., Kassandra, M., Godoy-Vitorino, F., Piceno, Y.M., Tom, L.M., Pantoja-Feliciano, I.G., Rivera-Rivera, M.J., Andersen, G.L. and Domínguez-Bello, M.G., 2012. Comparison of the fecal microbiota in feral and domestic goats. Genes3(1), pp.1-18.

O’Donnell, M.M., Harris, H.M., Ross, R.P. and O'Toole, P.W., 2017. Core fecal microbiota of domesticated herbivorous ruminant, hindgut fermenters, and monogastric animals. Microbiologyopen6(5), p.e00509.

Kubasova, T., Davidova-Gerzova, L., Babak, V., Cejkova, D., Montagne, L., Le-Floc'h, N. and Rychlik, I., 2018. Effects of host genetics and environmental conditions on fecal microbiota composition of pigs. PLoS One13(8), p.e0201901.

Correa-Fiz, F., Blanco-Fuertes, M., Navas, M.J., Lacasta, A., Bishop, R.P., Githaka, N., Onzere, C., Le Potier, M.F., Almagro-Delgado, V., Martinez, J. and Aragon, V., 2019. Comparative analysis of the fecal microbiota from different species of domesticated and wild suids. Scientific reports9(1), pp.1-15.

The authors thank the reviewer for suggesting the above references.

Other Comments:

  1. Figure 5 and associated text:  It would be helpful to the reader if the authors would perform a simple linear regression analysis on these data, deriving one equation for each kind of excreta (cow, goat, pig) and then comparing the regressions statistically (MANOVA, perhaps).  As it stands, we see the interaction of different excreta at different days.  However, a regression analysis would reveal whether one kind of excreta differs from another over the entire time period, i.e., which one gave a faster oviposition response (it might be that none of them did but we do not know that).

                                                The Figure 5 was replaced with a correlation analysis instead.

  1. The sentence that starts in Line 206 and runs until Line 208 does not make sense.  "It was observed that a significant difference between the mean larval mortality of Ae. albopictus and different vertebrate excreta (One-way ANOVA, F3,46= 31.68, P=0.00)."  Change this to read, "A significant difference between the mean larval mortality of Ae. albopictus and different vertebrate excreta was observed (One-way ANOVA, F3,46= 31.68, P=0.00)."

                                                Edited the sentence as advised.

  1. Again, Lines 229-231:  "Maximum number of total larval mortality was observed in the pig infusion and the minimum number of total larval mortality was observed from the cow excreta infusion (Figure 6)."  Change to read:  "Maximum larval mortality was observed in the pig infusion and minimum larval mortality was observed from the cow excreta infusion (Figure 6)."

                                                Edited the sentence as advised.

  1. Something else:  In all figure captions the authors repeat what software they used for data analysis.  The reader already was told in the Materials and Methods section - they do not need to be retold time and time again.

                                                The software names were repeated in the figures because the general norm is that the figures should stand alone themselves. However, if reviewers still think it is not necessary, the authors can omit the names accordingly in the next revision.

Round 2

Reviewer 1 Report

I thank the authors for considering the suggestions and improving their manuscript. I would recommend they add some minor improvements regarding the comments below:

L106- indicating the time of day is indicative only if we know when the night phase starts. Please prefer mentioning "x hours before night" instead of the exact time of beginning and end of observations.

L109- please add if the age of females changed over the batches of fresh females, or were they always 4 days-old?

2.6 and Point 17 of the authors response: I still find the method description unclear, please try to clarify. If I understand well: 1 cage contained 2 replicates of microcosms at different fermentation stages, all from the same animal excreta. There were therefore 3 cages in total, replicated 4 times.

Figure 7: please add the N (total number of larvae or pupae) used for each duration bar (this value could be added on the top or inside each bar). It seems important to add this value, as the durations are still strikingly short.

L326: please add reference of Muller book

Point 49 in the responses: there is no mention of transport before, when in the experimental design were the larvae transported?

Discussion: I would encourage the authors to discuss the fact that Aedes species can be found in brakish water, which justify their study; as opposed to usual artificial breeding sites reported for theses species elsewhere. ex reference to Ramasamy et al 2011 https://journals.plos.org/plosntds/article?id=10.1371/journal.pntd.0001369

Author Response

  1. L106- indicating the time of day is indicative only if we know when the night phase starts. Please prefer mentioning "x hours before night" instead of the exact time of beginning and end of observations.

                                                Added the mentioned part as advised.

  1. L109- please add if the age of females changed over the batches of fresh females, or were they always 4 days-old?

                                                They were always 4 days old belonging to a new batch of mosquitoes each day. Added the mentioned part as advised.

  1. 2.6 and Point 17 of the authors response: I still find the method description unclear, please try to clarify. If I understand well: 1 cage contained 2 replicates of microcosms at different fermentation stages, all from the same animal excreta. There were therefore 3 cages in total, replicated 4 times.

                                                Yes, the understanding of the reviewer is correct. However, attempts were made to clarify the meaning as per the modifications in L140-143.                                    

  1. Figure 7: please add the N (total number of larvae or pupae) used for each duration bar (this value could be added on the top or inside each bar). It seems important to add this value, as the durations are still strikingly short.

                                                An appropriate edit was done in L262-267indicating the total number of larvae used for different excreta type.

  1. L326: please add reference of Muller book.

                                                Added and edited all references including the Muller book.

  1. Point 49 in the responses: there is no mention of transport before, when in the experimental design were the larvae transported?

                                                What the authors really meant was taking the larvae from one place to another. So, the word “transport” is mistakenly used here. The correct word was edited in L385 as “ handling”.

  1. Discussion: I would encourage the authors to discuss the fact that Aedes species can be found in brakish water, which justify their study; as opposed to usual artificial breeding sites reported for these species elsewhere. ex reference to Ramasamy et al 2011 https://journals.plos.org/plosntds/article?id=10.1371/journal.pntd.0001369

Thank you for providing this reference. This was included in L64-66 and L284-285